# Way-Tu: A Framework for Tool Selection and Manipulation Using Waypoint Representations

**Abstract:** The ability to manipulate tools is essential for integrating intelligent robots in real-world settings, allowing them to significantly expand the range of tasks they can perform in daily life. To address this challenge, we introduce Way-TU, a novel framework that learns to generate waypoint representations (3D oriented keypoints) for motion planning in tool-use tasks. Our approach perceives the full environment, reasons over object geometry, and generates waypoints to guide the motion optimizer toward task completion, simultaneously enabling tool selection by identifying the most suitable tool among candidates. We evaluated our framework on three diverse tasks—minigolf, lifting, and hammering—and demonstrated a competitive manipulation performance compared to baselines, along with effective tool-selection capabilities.

**Keywords:** Learning Robot Fine Manipulation Skills, Tool Manipulation and Selection, Learning Waypoint Representations, Motion Optimization

## 1 Introduction

Our work aims to solve both tool manipulation and tool selection by explicitly considering the environment and adapting decisions according to the state of the task environment, rather than relying on predefined manipulation strategies. We propose a complete framework that perceives the full environment, identifies and interprets the objects within it, and generates waypoints through a trained network to guide a motion optimizer in planning feasible motions and selects the best tool between the candidates.

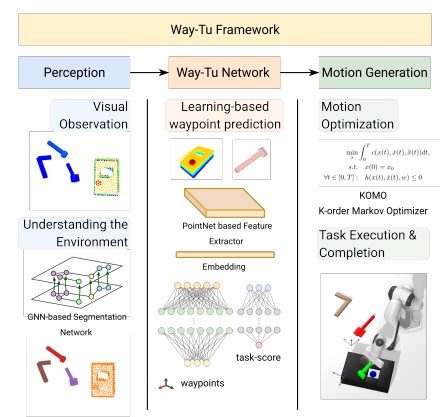

Figure 1: The framework integrates perception (point cloud segmentation), waypoint generation and score prediction, and motion optimization (KOMO) to find feasible solutions.

We augment a strong motion optimizer with supervised learning components that provide structured, high-level guidance. Rather than directly controlling the robot through learned policies—or relying solely on optimization to complete the task—our framework uses learning to infer task-relevant information, such as segmenting the scene, selecting the appropriate tool, and predicting waypoints as intermediate goals. This hybrid design combines the generalization and perceptual strengths of learning with the physical realism and constraint satisfaction offered by optimization. To this end, we propose Way-Tu, a framework that integrates perception, learning-based waypoint prediction, and motion optimization (Figure 1).

Our contributions are: (1) an end-to-end framework that jointly learns tool selection and waypoint generation, integrated with a motion optimizer for contact-rich tool use; (2) a generalizable data collection algorithm that produces diverse, valid samples across tasks without random exploration or manual annotation; and (3) a hybrid framework combining learning for generalization with motion optimization for physical feasibility, yielding a practical solution for tool use.

## 2  Related Work

Prior research has approached the problem from multiple perspectives, and the challenge of tool usage in robotics has been studied extensively [1, 2, 3] over the years. An increasing number of studies demonstrate that representing tools with sparse geometric structures, such as keypoints, is particularly effective for robotic manipulation [4]. Building on this idea, several studies have shown that robots can learn key aspects of tool manipulation through these sparse representations, enabling them to reason about tool affordances and functional parts rather than entire shapes. For example, KETO [2], GIFT [5], and ToolBot [6] leverage keypoint-based representations to learn the best ways to grasp the tool and manipulate it to complete the task. However, most of these works pay little attention to the environment or contact-rich aspect of the manipulation, and instead focus on a single tool placed on a table. In addition, most tool selection studies, whether aimed at choosing the right tool for a task [3] or reasoning about causal relationships between tools and tasks [7], have largely sidelined the manipulation process itself.

## 3  Simulation-Based Automated Waypoint Generation for Data Collection

We implemented a generalizable waypoint-generation algorithm that adapts to different tool-manipulation tasks by constraining grasp and interaction waypoints sampling based on object geometry and task definitions. For each sample, the environment is constructed by randomly generating both tool structures and a task platform, which are then placed on a table in random positions and orientations. The algorithm begins by selecting one of the available tools at random and isolating its point cloud from the environment. It then uses an antipodal grasp estimation algorithm to find all possible grasps. A valid antipodal pair is then randomly selected to define a grasp waypoint consisting of a position and a consistent orientation. Next, a contact point is chosen on the tool surface—deliberately positioned away from the grasp region. Using the target and contact point, the algorithm computes an initial manipulation waypoint aligned toward the task-specific target, followed by a goal waypoint representing the final task-achievement state. The goal waypoint is determined based on the task requirements and the optimal final position of the target object for successful task completion.

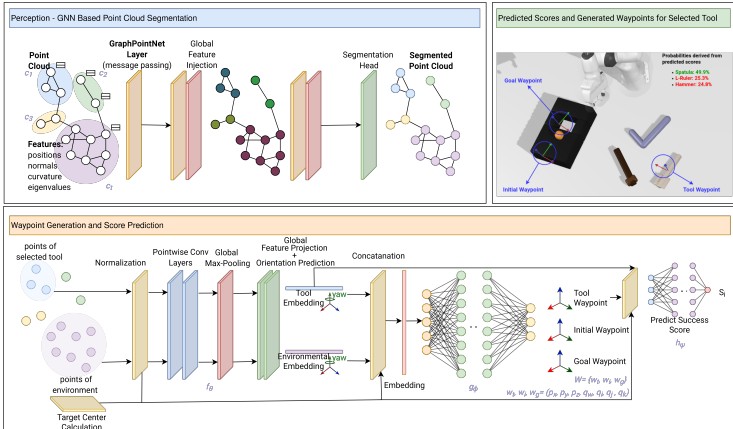

Figure 2: **Top-Left:** Graph PointNet-based segmentation of point clouds using geometric features. **Bottom:** Unified network for waypoint generation and score prediction using object embeddings extracted with a lightweight PointNet encoder. **Top-Right:** Example tool selection with predicted waypoints and chosen tool.

## 4 Methodology

Our framework sequentially handles perception, waypoint generation, and motion optimization to address both the manipulation and selection aspects of tool-usage problems. Learning-based components are integrated to understand the environment, identify the most suitable tool for the task, and support the motion optimizer during the manipulation phase.

For training the Way-Tu network, we first collected samples using our proposed data collection algorithm without any human interaction. Each sample contains the point cloud of the environment, the waypoints tested in the simulation, and a score representing the quality of task completion.

**GNN-Based Segmentation Module**  The segmentation module takes the raw environment point cloud and classifies each point into one of the tool or task-platform classes. The input point cloud is represented as a graph, where each point is a node with features including position, normal vector, curvature, and eigenvalues. The segmentation network (GraphPointNet) is composed of three message-passing layers, each followed by ReLU activation, and two global feature-injection layers. These are followed by an MLP segmentation head that assigns semantic labels to each point. This design enables the network to generalize to varying numbers of tools and to infer the current task platform without requiring explicit task labels.

**Feature Extractor**  For each object identified by the segmentation module (both tools and task-related platforms), we first normalize and scale their point clouds to reduce noise arising from random placements in the environment. The normalized point cloud is then passed to a lightweight PointNet-based encoder. This encoder applies two 1D convolutional layers with batch normalization and ReLU activations for point-wise feature extraction, followed by a global max-pooling layer to aggregate features across points into a compact embedding, and a fully connected layer for global feature projection. During training, the feature extractor is optimized with two objectives: (1) classifying the given object, and (2) predicting its orientation. We include orientation prediction because the unified network struggles more with estimating correct orientations for waypoints; explicitly learning orientation helps improve the quality of the embeddings.

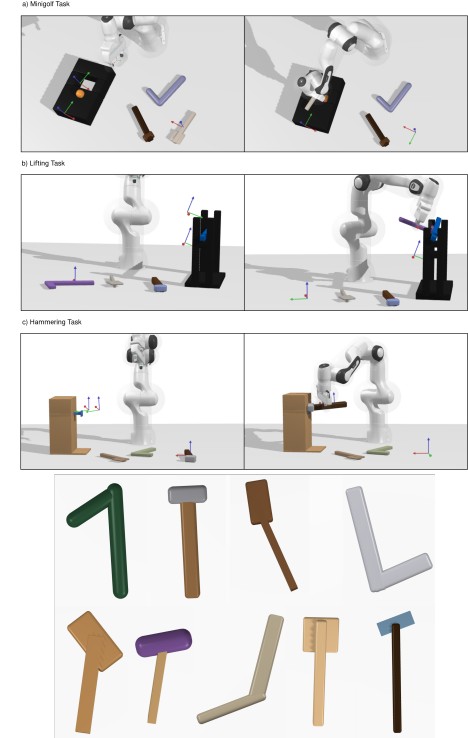

Figure 3: **Examples of tasks and tools.** Top: three tool-manipulation tasks (a) minigolf, (b) lifting, (c) hammering. Bottom: representative samples from hammer, spatula, and L-ruler families, illustrating geometric diversity.

**Unified Generator and Selector Module**  For each tool in the scene, the tool and environment embeddings produced by the feature extractor are concatenated with their normalization parameters, predicted yaw values, and the target object center (calculated from the environment point cloud using geometric methods). These inputs form a single embedding vector that is passed to the generator head, a deep MLP with normalization layers, residual connections, and SiLU activation functions, which predicts a set of three waypoints. The tool waypoint, together with the tool embedding and task encoding, is then passed to the selector network, a smaller MLP responsible for predicting a task success score. During training, the selector learns to evaluate the feasibility of each tool–waypoint combination. At inference time, it processes each tool individually and assigns a score reflecting the expected outcome if the corresponding waypoint set is used. The tool with the highest predicted score is selected as the optimal choice for the task. In practice, the ground-truth success score is

computed by jointly considering the stability of the tool grasp and the degree of task completion, providing consistent supervision for learning effective tool selection.

**Motion Generation and KOMO**  We employ the K-order Markov Optimizer (KOMO), which plans motions by formulating a nonlinear mathematical program with a sum-of-squares cost for improved regularization. KOMO is a trajectory optimization technique that can use a discrete set of waypoints as constraints to generate the motion path by minimizing the cost function. To achieve this minimization, equality constraints involving the generated waypoints are incorporated into the constraint function.

## 5  Experiments & Results

**Experimental Setup**  In our experiments, we consider three distinct tool-manipulation tasks, and we consider three tool families: hammer, spatula, and L-ruler (Figure 3). The **minigolf** task is a more goal-directed and challenging variation of a pushing task. The robot must push a ball, resting on an elevated platform, into a hole. In the **lifting** task, the robot must free a long stick-like target object trapped between thin vertical tubes by applying an upward force. In the **hammering** task, the goal is to drive a nail—partially embedded in a small ball—into a wall.

**Tool Selection Evaluation**  To evaluate the performance of the tool selection module, we measured the ratio of tools chosen during the manipulation experiments Table 1. In the data collection phase, the tool was selected randomly, resulting in Data Collection (DC) having an almost uniform distribution over tools. The selection module learns to jointly map tools, tasks, and grasping

Table 1: Tool selection rates (%).

| Tool | DC | | | Way-Tu | | |
|------|------|------|------|------|------|------|
| | Mini | Lift | Hamm | Mini | Lift | Hamm |
| L-ruler | 32.7 | 31.1 | 34.5 | 25.0 | **81.8** | 34.78 |
| Spatula | 36.3 | 37.4 | 35.5 | **53.6** | 18.2 | 13.04 |
| Hammer | 31.0 | 31.0 | 29.9 | 21.4 | 0.0 | **52.2** |

positions to their resulting performance. Consequently, at inference time, the module tends to select tools that it has internally associated with higher success probabilities, rather than following the random distribution present in the training data.

**Manipulation Performance and Baseline Comparisons**  The success rates of our model and different baselines can be seen in Table 2. We evaluate a motion-optimizer-only baseline using the KOMO framework without any additional intermediate goals. Even with multiple randomized starting configurations per environment, the pure motion optimizer failed in all three tasks. Way-Tu-DC demon-

Table 2: Success rates (%) across tasks.

| Method | Mini | Lift | Hamm |
|--------|------|------|------|
| Pure KOMO | 0.0 | 0.0 | 0.0 |
| Way-Tu-DC | 45.3 | 40.9 | 35.3 |
| KETO | 33.4 | 41.7 | 22.3 |
| ToolBot | 16.7 | 33.4 | 23.5 |
| Way-Tu | **75.0** | **77.8** | **69.6** |

strates the performance of our data collection algorithm, which augments the KOMO motion optimizer with a heuristic that introduces feasible waypoints. Unlike pure KOMO, the data collection algorithm was able to solve all tasks, to some extent. We selected KETO [2] and ToolBot [6] for learning-based baselines. Both models were originally designed to operate only on the tool; we adapted them to also learn from the environment. When the environment point cloud was included, the predicted keypoints became unstable and less consistent. Compared with all baselines, Way-Tu achieved the highest success rates across all three tasks, demonstrating the effectiveness of our method in jointly considering environment point clouds and tools while generating complete waypoint sets with orientations.

## 6  Conclusion

In this study, we proposed Way-Tu, an end-to-end framework that jointly addresses tool manipulation and tool selection by considering the full environment rather than focusing solely on the tool. We validated our framework on three diverse tool-manipulation tasks—*minigolf*, *lifting*, and *hammering*. Across all settings, Way-Tu achieved competitive manipulation performance compared to relevant baselines, while also providing reliable and meaningful tool-selection results.

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
