# OpenReview forum: "Way-Tu: A Framework for Tool Selection and Manipulation Using Waypoint Representations"
_robot-learning.org/CoRL/2025/Workshop/Dexterous_Manipulation — CoRL 2025 Workshop Dexterous Manipulation Spotlight_

### Official Review · Reviewer_oMaS · 2025-09-10
**Review of Way-Tu: A Framework for Tool Selection and Manipulation Using Waypoint Representations**

**Rating:** 7
**Confidence:** 3

**Review:**

- Brief summary
This paper introduces Way-Tu, a framework that integrates perception, waypoint generation, and motion optimization to jointly solve tool selection and tool manipulation tasks. The work combines a learning-based segmentation and waypoint generation module with a motion optimizer (KOMO), and demonstrates its effectiveness on three representative tasks—minigolf, lifting, and hammering—showing clear improvements over strong baselines. The approach is novel in its joint treatment of tool selection and manipulation while leveraging structured waypoint representations.

- Review
The paper tackles an important problem in robotic manipulation: enabling robots to flexibly select and use tools in contact-rich environments. The proposed Way-Tu framework is well-motivated and contributes three key aspects: (1) an end-to-end pipeline that unifies perception, tool selection, and waypoint prediction, (2) a generalizable automated data collection procedure, and (3) integration with trajectory optimization for physically feasible execution.

- Strengths
Clear contribution: The hybrid design—learning for high-level reasoning and optimization for low-level feasibility—is compelling and well-positioned within the literature.
Novelty: Unlike prior works (e.g., KETO, ToolBot), the method explicitly reasons about the environment rather than focusing only on the tool, which broadens applicability.
Experimental results: Strong performance across three diverse tasks with meaningful comparisons to baselines demonstrates the effectiveness of the approach. The tool selection results (Table 1) and task success rates (Table 2) are convincing.
Clarity: The paper is generally well-structured and readable, with a clear separation between methodology, data generation, and evaluation.

- Weaknesses / Limitations
Limited scope of tasks: Only three tasks (minigolf, lifting, hammering) are tested. While diverse, they are still simulation-based and relatively constrained. Real-world deployment or additional task families would strengthen the claims of generalization.
Ablation analysis: The paper lacks deeper ablations on the design choices (e.g., the importance of orientation prediction in the feature extractor, or the role of residual connections in the generator). Such analysis could clarify which parts of the pipeline drive performance.
Clarity of comparisons: While KETO and ToolBot are adapted for environmental input, the details of their adaptation are brief. More discussion on the fairness of the comparison would improve rigor.
Scalability: The framework depends on precise point cloud segmentation and KOMO optimization, which may limit efficiency in more complex or cluttered real-world settings.

- Significance
The work is a valuable step toward a generalizable tool-use in robotics. By unifying selection and manipulation through waypoint-guided optimization, it offers a practical and conceptually clean approach. While the scope of evaluation is somewhat narrow, the proposed ideas are original and of interest to the CoRL workshop community.

---

### Official Review · Reviewer_hGjN · 2025-09-10
**Review for Submission Number 22**

**Rating:** 6
**Confidence:** 3

**Review:**

### [Summary]
They introduce Way-TU, a novel framework that generates waypoint representations (3D oriented keypoints) for motion planning in tool-use tasks. The framework jointly addresses tool selection and manipulation by integrating perception, learning, and motion optimization.
### [Strengths]
The system provides a framework that can select and manipulate tools in unstructured scenes. It also enables end-to-end execution of tasks with previously unseen tools by predicting suitable waypoints and motions.
### [Weaknesses]
The figures are difficult to interpret due to small script size, and the paper could better explain the role of each module beyond just architectural details. Moreover, grasp waypoint stability is not discussed, raising questions about applicability to real-world cases where weight distribution and CoM matter (e.g., hammers). The use of a two-finger gripper limits the connection to dexterous manipulation, as stability and multi-contact control are not fully considered.
### [Overall Feedback]
While the direct connection to dexterous manipulation workshops is limited, the paper presents a meaningful contribution toward hybrid learning-control approaches for tool-use. Its strengths in unstructured environments and unseen tool generalization make it a valuable piece of work.

---

### Decision · Program_Chairs · 2025-09-18

Accept (Spotlight)